# Mechanistic Insights on Metformin and Arginine Implementation as Repurposed Drugs in Glioblastoma Treatment

**DOI:** 10.3390/ijms25179460

**Published:** 2024-08-30

**Authors:** Anna-Maria Barciszewska, Agnieszka Belter, Jakub F. Barciszewski, Iwona Gawrońska, Małgorzata Giel-Pietraszuk, Mirosława Z. Naskręt-Barciszewska

**Affiliations:** 1Intraoperative Imaging Unit, Chair and Department of Neurosurgery and Neurotraumatology, Karol Marcinkowski University of Medical Sciences, Przybyszewskiego 49, 60-355 Poznan, Poland; ambarciszewska@ump.edu.pl; 2Department of Neurosurgery and Neurotraumatology, University Clinical Hospital, Przybyszewskiego 49, 60-355 Poznan, Poland; 3Institute of Bioorganic Chemistry of the Polish Academy of Sciences, Noskowskiego 12, 61-704 Poznan, Poland; abelter@ibch.poznan.pl (A.B.); jbarciszewski@ibch.poznan.pl (J.F.B.); iwonag@ibch.poznan.pl (I.G.); giel@ibch.poznan.pl (M.G.-P.)

**Keywords:** metformin, arginine, temozolomide, glioblastoma, 5-methylcytosine, 8-oxo-deoxyguanosine, DNA methylation, epigenetics

## Abstract

As the most common and aggressive primary malignant brain tumor, glioblastoma is still lacking a satisfactory curative approach. The standard management consisting of gross total resection followed by radiotherapy and chemotherapy with temozolomide only prolongs patients’ life moderately. In recent years, many therapeutics have failed to give a breakthrough in GBM treatment. In the search for new treatment solutions, we became interested in the repurposing of existing medicines, which have established safety profiles. We focused on the possible implementation of well-known drugs, metformin, and arginine. Metformin is widely used in diabetes treatment, but arginine is mainly a cardiovascular protective drug. We evaluated the effects of metformin and arginine on total DNA methylation, as well as the oxidative stress evoked by treatment with those agents. In glioblastoma cell lines, a decrease in 5-methylcytosine contents was observed with increasing drug concentration. When combined with temozolomide, both guanidines parallelly increased DNA methylation and decreased 8-oxo-deoxyguanosine contents. These effects can be explained by specific interactions of the guanidine group with m^5^CpG dinucleotide. We showed that metformin and arginine act on the epigenetic level, influencing the foreground and potent DNA regulatory mechanisms. Therefore, they can be used separately or in combination with temozolomide, in various stages of disease, depending on desired treatment effects.

## 1. Introduction

Glioblastoma, a diffuse glial brain tumor, is the most common primary malignancy in the adult central nervous system (CNS) and is also the most aggressive and lethal entity [1,2]. It invades the adjacent brain tissue and is characterized by uncontrolled proliferation, active angiogenesis, invasiveness, and the ability to avoid apoptosis. The median survival is only approximately 14.6 months [2,3]. The median progression-free survival is 5.3 months, and the 5-year survival rate is up to 4% [4,5]. The vast majority of patients will die within 2 years from the time of diagnosis due to the disease relapse [4]. The currently used therapeutic strategies do not offer good curative solutions, allowing only a prolongation of and improvement in the patient’s quality of life. A complete resection of the tumor is not possible due to its diffuse nature. The up-to-date standard management of glioblastoma includes gross total surgical resection followed by radiotherapy in combination with chemotherapy with temozolomide (TMZ) [6]. TMZ therapy increases life expectancy by 2 months and increases 2-year survival by up to 26.5% [7,8]. Providing a breakthrough in 2005 [3], TMZ remains a better-than-others drug. However, it is still not curative. Its effectiveness is limited by resistance mechanisms [9]. Therefore, there is constantly an urgent need for novel, more effective therapeutic approaches. One of the cost-effective strategies for this is drug repurposing. It can accelerate time-consuming translational research by identifying new implementations for well-known and tested therapeutics.

The basis for such an approach is the adequate characterization of the molecular alterations in gliomagenesis to identify substances that can interfere with those events. Moreover, the currently used chemotherapeutics should be more precisely characterized regarding their mechanism of action to find new potential targets. It is widely described that the first-line medication in GBM, TMZ, shows cytotoxicity with the addition of methyl groups at N7 (>70%) and O6 (6%) sites of guanine and N3 (9%) sites of adenine in genomic DNA. Interestingly, the smallest part of modified bases, O6-methylguanine (O6-MeG), is regarded as critical for TMZ-induced cytotoxicity [10]. It is the promoter methylation of the MGMT (O6-methylguanine-DNA methyltransferase) DNA-repair gene, resulting in its silencing, which has been associated with longer survival in patients with glioblastoma who receive alkylating agents [11]. However, with much available data, the conclusion of withholding TMZ in patients with unmethylated MGMT cannot be made yet [12,13]. Recently we showed different mechanisms of temozolomide action in glioblastoma through the modulation of global DNA methylation [14,15].

DNA methylation is a part of the immense epigenetic machinery that regulates gene expression without changing the nucleotide sequence. Those processes, driven by intrinsic and external factors, are important for the pathogenesis of many diseases, including cancer [16,17]. Understanding the mechanism of these modifications is crucial for the evaluation of the cancer formation processes and further drug development. The methylation of cytosine residues in DNA is a dominant epigenetic modification of the mammalian genome, determining the site and time of gene expression or its silencing [18,19]. The appropriate level of 5-methylcytosine (m^5^C) at a specific time and place determines the development and differentiation of a given cell [20]. The mechanism involves the covalent binding of a methyl group (-CH_3_) to the fifth carbon of cytosine in CpG dinucleotides where S-adenosylmethionine (SAM) is the methyl donor. The reaction is catalyzed by DNA methyltransferases [18]. The high stability of the C-C covalent bond of 5-methylcytosine and the lack of specific demethylases make removing the methyl group (demethylation) unique and specific. It can take place through oxidative demethylation catalyzed with Ten-Eleven Translocation (TET) enzymes, which are oxo-reductases, or through spontaneous oxidation and methyl group removal as a result of the action of reactive oxygen species (ROS) [21]. ROS are the main factor damaging cell components including DNA [22]. Under physiological conditions, the emerging DNA damage is repaired [23], but the imbalance between ROS and free radical scavenging compounds in cancer cells can induce cellular stress [24]. Increased ROS levels may lead to mutations, genome instability, a higher proliferation rate, resistance to chemotherapy, or increased glucose metabolism [25,26]. Cancer cells show elevated levels of 8-oxo-7,8 dihydro-2-deoxyguanosine (8-oxo-dG), a product of oxidative DNA damage. It is formed due to the chemical modification of the guanine moiety with the hydroxyl radical (•OH) [27], which is the most reactive ROS with a very short half-life (10^9^ s–10^8^ s). In addition to Watson–Crick bond formation with cytosine, 8-oxo-dG mispairs adenine and induces G-T transversions [28].

The complete or partial loss of 5-methylcytosine (hypomethylation) occurs in many human neoplasms [29], and GBM is linked to higher malignancy, tumor invasion, and proliferation [30,31,32]. Hypomethylation is a direct result of the increase in oxidative stress within the cell [33]. Therefore, not necessarily hyper- but eumethylation should be the desired therapy goal. This perspective is very attractive because, unlike genetic alterations, DNA methylation is potentially modulated by pharmacological intervention and comprises an attractive anticancer agent [34].

The mainstream therapy concept from an epigenetic perspective is the induction of global DNA hypomethylation to reprogram tumor cells to a normal-like state by affecting multiple pathways and sensitizing them to chemotherapy and immunotherapy [35]. While the drugs increase oxidative stress, which intensifies hypomethylation [36,37,38], DNA hypermethylation, observed in the course of treatment, is therefore regarded as an event leading to therapy resistance [35,39]. However, the perfect solution is probably not one way and needs a balance in oxidative stress generation, as well as the recognition of healthy cells outside the treated cancer [36,40].

Our project aimed to evaluate possible drugs to be repurposed into GBM treatment in the context of changes in global DNA methylation and oxidative state, monitored through the m^5^C and 8-oxo-dG levels, respectively. Those two parameters could be introduced in the estimation of the effects of treatment in the clinical setting. As tested drugs, we included two guanidine derivatives, metformin and arginine (Figure 1). TMZ served as a reference and concomitantly used substance.

Metformin (MF) is a biguanide drug widely used in the treatment of type 2 diabetes (T2D). It is a low-molecular-weight amphoteric compound and has a non-polar carbon chain, which gives it lipophilic properties (log P = −2.6) [41]. Metformin is one of the few drugs that can cross the blood–brain barrier (BBB) [42,43]. MF inhibits gluconeogenesis in the liver, sensitizes peripheral cells to insulin, increases glucose uptake, inhibits mitochondrial respiration, and reduces glucose absorption in the gastrointestinal tract [44,45,46]. The molecular anticancer mechanism of metformin is unknown. However, it is suggested that MF affects mitochondria through oxidative stress or by regulating the activity of the AMPK pathway [47]. The lack of tumor progression in GBM due to MF has been suggested [48,49,50,51].

Temozolomide (TMZ) shows lipophilic properties (Log P = −1.153), which enable it to cross the BBB and is therefore available in the CNS [52]. It has been shown that glioma cells treated with TMZ and MF showed a significant reduction in the proliferation rate compared to cells treated with TMZ alone [41]. One can assume that metformin increases the effectiveness of TMZ and may be used in cancer therapy.

Arginine (Arg) is an amino acid that also contains the guanidine group (log P = −4.2). It is not essential in non-cancerous human cells but is crucial for the survival of cancer cells. Defective arginine synthesis is one of cancer’s most common metabolic weaknesses [53]. Cancer cells often alter mitochondrial function, and arginine starvation has been shown to damage mitochondria, causing an increased accumulation of ROS and subsequent genome instability [54,55,56]. Arginine supports several metabolic reactions, including the synthesis of nitric oxide, polyamines, glutamine, and proline, all of which are important cell growth and survival regulators. Preventing arginine degradation may improve the antitumor effects of T and NK cells [57,58,59]. It is known that in contrast to humans where the body can synthesize arginine, certain cancer cells are auxotrophic for arginine, which means that its synthesis in the cell may be insufficient. Even more, this observation was the basis for the development of new approaches to some cancer treatments. The analysis of the cellular effects of arginine and metformin on a molecular level is very intriguing. It has been found that metformin shows epigenetic potential to inhibit the growth of cancer cells, and arginine executes a similar effect. There is a structural analogy of metformin (biguanide) with arginine (single guanidine group).

Arginine plays an important role in cancer progression and the process of antitumor immunity, but its metabolism significantly impacts the immunological characteristics of tumors. Cancer cells often depend on exogenous arginine to meet their needs because they lose the ability to synthesize arginine intracellularly due to the loss of a key arginine-producing enzyme, argininosuccinate synthetase [60].

## 2. Results

### 2.1. Cytotoxic Effect of Metformin, Arginine, and Temozolomide in Neoplastic and Normal Cell Lines

Cell viability was determined using the MTT assay for tested compounds temozolomide (TMZ), metformin (MF), and arginine (Arg) at concentrations of 1, 8, 63, 250, and 2000 µM (0–3.3 on a logarithmic scale). One can see that temozolomide above 50 µM (1.7 on a logarithmic scale) strongly inhibits the growth of all cell lines. The tested cell lines are most sensitive to TMZ. MF and Arg in the studied concentration range did not show cytotoxic effects on T98G, U138, or HaCaT cells. As it is shown below, TMZ is the most harmful of all the compounds analyzed in the study (Figure 2).

### 2.2. Cytotoxic Effect of a Combination of Metformin, and Arginine with Temozolomide in Glioblastoma Cell Line

For these studies, we selected the T98G cell line because it is the best equivalent to glioblastoma according to its current definition carrying IDHwt phenotype, as well as being more aggressive and treatment-resistant regarding MGMT promoter methylation [1,61]. The cell viability of the T98G cell line subjected to combination therapy was determined using the MTT assay with MF and Arg in concentrations of 1, 8, 63, 250, and 2000 μM (0–3.3 on a logarithmic scale) and TMZ of 0, 1, 30, and 100 μM. The effectiveness of the combination treatment with TMZ and MF and TMZ and Arg on T98G cells is presented in Figure 3. Temozolomide shows reduced toxicity in the presence of arginine (Figure 3B), but the composition of metformin with temozolomide at concentrations above 1 µM TMZ inhibits the growth of the T98G cell line (Figure 3A). One can conclude that both MF and Arg act antagonistically with TMZ and reduce its toxicity.

### 2.3. The Effect of Metformin and Arginine on Genomic DNA Methylation in Cell Lines

The effects of MF and Arg on HaCaT, T98G, and U138 cell lines on the global level of m^5^C in cellular DNA were analyzed. We used two different glioblastoma cell lines and low drug concentrations. The global DNA methylation level (expressed as R) was analyzed after treatment with 0–1000 µM MF and 0–750 µM Arg for 3, 24, and 48 h (Figure 4A,B). Concentration and time dependent changes in m^5^C content were observed. No significant changes in DNA methylation were observed for non-cancer cell lines treated with MF. Both glioblastoma cell lines showed a similar methylation pattern, different from that of normal cell lines. Clear demethylation was observed for cancer cell lines (T98G and U138) with increasing MF concentration and incubation times (Figure 4A). T98G and U138 cell lines treated with increasing arginine concentration and at longer incubation times of 24 and 48 h showed decreasing methylation, similar to MF. Interestingly, for a short incubation time for cancer cell lines, hypermethylation was observed, similar to HaCaT (Figure 4B). This confirms earlier observations showing that MF is more toxic than Arg (Figure 3).

### 2.4. Quantification Estimation of m^5^C and 8-oxo-dG in DNA of T98G Cell Line Treated with Metformin, Arginine and Temozolomide

The amount of m^5^C and 8-oxo-dG residues in genomic DNA (Figure 4C,D) of the T98G cell line exposed to TMZ, MF, Arg, and their combination was assessed. One can see that treatments with MF and Arg in the presence of TMZ cause an increase in the m^5^C amount from 3 × 10^6^ to 1 × 10^6^ residues per genome (Figure 4C). However, the highest TMZ and MF or Arg concentrations induce oxidative demethylation and a reduction in m^5^C from 17 × 10^6^ to 5 × 10^6^ residues. Interestingly these compounds do not increase the amount of 8-oxo-dG in DNA (Figure 4D). The decrease in the amount from 3.5 × 10^6^ to 2.5 × 10^6^ residues of 8-oxo-dG suggests the inhibition of the reactivity of guanine C-8 towards oxidation with guanidine compounds. Changes in the amount of m^5^C and 8-oxo-dG levels expressed as the number of residues per genome allow for a better insight into functional and structural considerations.

### 2.5. The Effect of Metformin, Arginine, and Temozolomide on Genomic DNA Methylation

We analyzed the combined effects of MF (0, 1,10, 50, and 100µM) and Arg (0, 100, 250, 500, and 750µM) with TMZ (0, 1, 30, and 100 µM) for 24 h on glioblastoma cell lines (T98G, U138) and keratinocyte cell lines (HaCaT) (Figure 5). The effect of MF and Arg in combination with TMZ on glioma cell lines is similar, although the T98G cell line is more sensitive. One can see that MF and Arg cause a concentration-dependent increase in genomic DNA methylation in all cell lines, even in the presence of TMZ.

## 3. Discussion

Despite the various available treatments and advances in chemotherapy, the prognosis for people diagnosed with glioblastoma is still poor. Current treatment includes maximal surgical resection, radiotherapy, and chemotherapy with TMZ. TMZ is standard chemotherapy for GBM, but it is not very effective due to tumor resistance. Therefore, one should overcome these obstacles by developing new, more effective strategies against glioblastoma. Recently, great interest in proven drugs for various diseases that may gain new applications has been observed. Repurposed drugs should provide benefits for cancer patients, despite being developed for an unrelated disorder. Repurposing may be critical for rare cancers where drug development is very costly. It has been demonstrated that MF (Figure 1), which is chemically dimethyl biguanide, showed antitumor activities [62]. The treatment of diabetic patients with metformin reduces the risk of concomitant malignancies. It facilitates cancer cell death through several signaling pathways, although molecular mechanisms underlying the antineoplastic properties of metformin remain elusive [63]. Metformin is characterized by five unusual N atoms and electron delocalization over the nitrogen atoms [64]. Various data indicate that metformin can prevent carcinogenesis and suppress tumor growth and metastasis [65]. MF can protect normal cells against the genotoxic effects of carcinogen agents and can boost DNA repair and ROS generation by mitochondria [66,67]. It has been shown that MF reduces the motility and invasiveness of both endometrial and ovarian cancer cells partly due to the decreasing synthesis of H19 RNA [68]. This long noncoding RNA alters DNA methylation genome-wide due to its binding and inhibiting S-adenosylhomocysteine hydrolase [68]. These data suggested that MF may alter gene methylation in malignant cells [69] through DNA damage and oxidative stress, leading to the activation of various epigenetic mechanisms of cell death [67,70]. For our study, we also selected arginine as a metformin analog.

The structure of the Arg molecule (Figure 1 and Figure 6A) suggests its participation in a large number of cellular processes. It can bind to the phosphate anion and therefore is involved in the catalysis of phosphorylation reactions [71]. Arg also participates in maintaining the charge of many proteins [72] and inhibits the self-splicing of the Tetrahymena intron by competing with guanosine. It has been proposed that the guanidine group of arginine can substitute for the two donor groups of guanine [73].

Metformin and arginine both contain the guanidine group (Figure 6A), which has a strong affinity to nucleic acids. The arginine side chain mimics the hydrogen-binding face of guanidine. Its positive charge allows it to participate in interactions with the phosphate backbone of DNA that are not available to other amino acids. To see their impact on glioblastoma, we analyzed global changes in DNA methylation (m^5^C), an epigenetic marker, as well as of 8-oxo-dG, a marker of general oxidative damage. The properties and functions of these two DNA constituents are well known and can be used as probes to monitor various cellular processes. This can be carried out because both drugs affect the levels of both markers in the cell differently.

Now, the question is how MF and Arg recognize and interact with methylated CpG dinucleotides, m^5^CpG, and what the mechanism of that process is. For a deeper understanding of the cellular effects of these drugs, we also used temozolomide, a methylating agent, which is a first-line drug in glioblastoma chemotherapy.

Our experiments showed that temozolomide separately is more toxic than metformin and arginine (Figure 2). We did not see the harmful effect of both of them up to 2 mM concentrations. m^5^C in the DNA of normal cells is on the same level after treatment with metformin. However, metformin and arginine induce the demethylation of m^5^C in cancer cells only after longer treatment, e.g., 24 h. Interestingly, arginine stimulates DNA methylation in normal and cancer cells, with the exceptions of T98G and U138 for longer incubation times. One can observe that metformin, arginine, and temozolomide stimulate the hypermethylation of DNA (Figure 4 and Figure 5). To explain this observation on the molecular level we looked for structural similarities between guanidine compounds and their involvement in interactions with m^5^CpG dinucleotides. 5-methylcytosine can be oxidized to 5-hydroxymethylcytosine but the C-8 of guanosine is oxidized to 8-oxo-dG. Generally, the binding of ligands to a specific target depends on several factors such as shape, charge, hydrogen bonding capacity, flexibility, planarity polarity, and size. If guanidine compounds (MF and Arg) are supposed to bind nucleic acids, they have to be comparable at both the structural and physicochemical levels.

Our results suggest that metformin and other guanidines can bind to DNA. The electronic absorption bands of metformin in a complex with DNA proved an intercalating mode of interaction [74]. The circular dichroism data supports the idea that metformin does not intercalate into the DNA helix, but the induced circular dichroism (ICD) of metformin in the presence of DNA suggests its binding to the helical structure of DNA. This binding varies with the base pair ratios and preferred AT-rich domain over GC-rich ones [74].

It has been found that 5-methylcytosine can be recognized by the amino acid side chain of specific DNA-binding proteins [75]. Proteins in the complex with methylated DNA reveal common recognition modes for m^5^C. It involves an m^5^C-Arg-G motif, where the arginine side chain forms hydrogen bonds with the guanine 3′ of m^5^C in a CpG step via Hoogsteen hydrogen bonding interactions with O6 and N7 atoms. Furthermore, a structural analysis of the complexes shows that the methyl group can form van der Waals contacts with the guanidine moiety and influence the reactivity of carbon 5 of cytosine towards methylation [75]. The importance of these interactions has been supported by observations that *ZFP57* mutations, found in patients with transient neonatal diabetes, which change base interactions, lead to abolished DNA binding activity. If so, the absence of a specific protein at the binding site might provide room for another protein to compete and modify the m^5^C group [75].

The newly proposed mechanism of MF and Arg action is based on the involvement of oxygen 6 and nitrogen 7 of guanine of m^5^CpG dinucleotides in forming Hoogsteen hydrogen bonds with the guanidine motif (Figure 6B). This explains the reactivity and the lower toxicity of TMZ. The presence of metformin and arginine suggests that the binding of the guanidine group with the Hoogsteen edge of guanine oxygen 6 and nitrogen 7 makes the methylation of DNA impossible because of its involvement in hydrogen bonding. The entanglement of N7 in the hydrogen bond also diminished carbon 8 reactivity towards oxidation and the formation of 8-oxo-guanine (Figure 6B) [75,76,77]. The observed decreasing level of the oxidative stress marker suggests the protective effect of metformin and arginine on the guanine moiety (Figure 6). In this condition, the reactivity of carbon 8 of the guanine moiety is less prone to a radical oxidation reaction because of the guanidine fork interaction with the major groove edge of guanine of the m^5^CpG dinucleotide.

Additionally, a guanidine can bind to the phosphodiester group of the DNA chain (Figure 6B). Metformin and arginine having a guanidine group perfectly fulfill all structural requirements to interact with DNA and form a specific regulatory network of interactions. One can say that upon using DNA methylation as a probe for functional studies, we can propose a new mechanism for the metformin action of repurposed drugs that fits very well with the observed experimental results.

## 4. Materials and Methods

### 4.1. Chemicals and Reagents

Metformin (MF) and Arginine (Arg) (Merck, Darmstadt, Germany) stock solution of 10 mM in water was used to prepare the required concentration with a complete medium. Temozolomide (Merck, Darmstadt, Germany) was dissolved in dimethyl sulfoxide (DMSO, Sigma/Merck, Darmstadt, Germany). MF (1.38 mg/mL) and Arg (2.28 mg/mL) are soluble in water, but TMZ is soluble in an organic solvent, DMSO, and therefore, it is used for control experiments. [γ-P^32^] ATP (6000 Ci/mmol) was purchased from Hartmann Analytic GmbH (Braunschweig, Germany), T4 polynucleotide kinase from USB (Thermo Fisher Scientific, Waltham, MA, USA), and micrococcal nuclease, spleen phosphodiesterase II, apyrase, RNase P1, thiazolyl blue tetrazolium bromide, inorganic salts, cellulose plates and methanol from Sigma/Merck (Darmstadt, Germany), and the Genomic Mini kit for DNA isolation was supplied by A&A Biotechnology (Gdańsk, Poland).

### 4.2. Cell Line and Culture Conditions

Human glioblastoma (T98G, U138), and human epidermal keratinocyte (HaCaT) cell lines were purchased from ATCC (Manassas, VA, USA). Cell lines T98G and U138 were cultured in EMEM medium from ATCC (USA) and HaCaT in EMEM (Sigma). Each medium was supplemented with 10% (*v*/*v*) fetal bovine serum (FBS, Sigma/Merck) and 10 mg/mL antibiotics (penicillin 100 U/mL 330 and streptomycin 100 µg/mL) from ATCC (Manassas, VA, USA). Cells were cultured at 37 °C with 5% CO_2_ in humidified air. After 24 h, cells were washed with phosphate-buffered saline (PBS, Merck), placed in a fresh medium, and treated with metformin, arginine, and temozolomide alone or in a designed combination.

### 4.3. Cell Viability Assay

Cell viability was evaluated with a dye-staining method, using 3-(4,5-dimethyl-2-thiazolyl)-2,5-diphenyl-2Htetrazolium bromide (MTT). Cell lines (HaCaT, T98G, and U138) were seeded in 96-well culture plates at a density of 1 × 10^4^ cells/well and grown in the supplemented medium at 37 °C under a 5% CO_2_ atmosphere. The cell lines were then treated with MF, Arg, and TMZ at selected concentrations. After 24 h, the supernatant was washed out, and 100 µL of MTT solution in medium (0.5 mg/mL final concentration of MTT) was added to each well for 2 h. After the incubation, the unreacted dye was removed through aspiration. The formazan crystals were dissolved in 100 µL/well DMSO and measured spectrophotometrically in a multi-well Synergy2 plate reader (BioTek Instruments, Winooski, VT, USA) at a test wavelength of 492 nm and a reference wavelength of 690 nm. Values represent the means ± SD from at least four independent experiments.

### 4.4. Treatment of Cell Lines with Metformin and Arginine

MF and Arg stock solution were added directly to the culture medium (with 90–95% confluence) to obtain different concentrations (1–2000 µM) and incubated for 3, 24, and 48 h. For the control, the cells were treated with H_2_O only. After 3–48 h of MF and Arg treatment, cells were washed with PBS, trypsinized, and collected by centrifugation at 4000 rpm for 10 min. The cellular pellets were quickly frozen and stored at −20 °C for DNA isolation.

### 4.5. Treatment of Cell Lines with the Combination of Metformin, Arginine, and Temozolomide

MF, Arg, and TMZ stock solutions and their dilutions were added directly to the culture medium to achieve the designed concentration. In experiments with TMZ, the final DMSO concentration in each cell culture was 0.8%. Cell cultures (with 90–95% confluence) were washed with PBS placed in fresh medium and treated with MF and Arg only (1, 8, 63, 250, 2000 μM), TMZ only (1, 30, or 100 μM), and their combination. The control cells were treated with H_2_O (for MF and Arg) and DMSO (for TMZ). After 24 h incubation, the cells were washed with PBS, trypsinized, and collected by centrifugation at 4000 rpm for 5 min. The cellular pellets were quickly frozen and stored at −20 °C for DNA isolation.

### 4.6. Isolation of the DNA

Cells were first detached from the culture plate using a trypsin-EDTA solution and centrifuged at 1200 rpm for 5 min at 4 °C. The cell pellet was washed in PBS and then processed for DNA isolation using a genomic DNA extraction kit (A&A Biotechnology), according to the manufacturer’s protocol. The purity and concentration of DNA preparations were checked by measuring UV absorbance at 260 and 280 nm. The A260/A280 ratios were between 2.0 and 2.1. DNA was stored at −20°C for further analysis.

### 4.7. DNA Hydrolysis, Postlabeling and TLC

DNA (dried, 1 μg) was dissolved in succinate buffer (pH 6.0) containing 10 mM CaCl_2_ and digested with 0.001 units of spleen phosphodiesterase II and 0.02 units of micrococcal nuclease for 5 h at 37 °C. Moreover, 0.17 μg of DNA digest was labeled with 1μCi [γ-^32^P]ATP (6000 Ci/mmol, Hartmann Analytic GmbH) and 1.5 units of T4 polynucleotide kinase (USB, UK) in 10 mM bicine-NaOH pH 9.7 buffer containing 10 mM MgCl_2_, 10 mM DTT, and 1 mM spermidine. After 0.5 h at 37 °C, apyrase (10 units/mL) in the same buffer was added and incubated for another 0.5 h. The 3′nucleotide phosphate was cleaved off with 0.2 μg RNase P1 in 500 mM ammonium acetate buffer, pH 4.5. The identification of [γ-^32^P]m^5^dC was performed with two-dimensional thin-layer chromatography (TLC) on cellulose plates using a solvent system: isobutyric acid/NH_4_OH:H_2_O (66:1:17 *v/v*) in the first dimension and 0.2 M sodium phosphate (pH 6.8)-ammonium sulfate-n-propyl alcohol (100 mL/60 g/2 mL) in the second dimension. Radioactive spot analysis was conducted with the Phosphoimager Typhoon Screen (Pharmacia, Uppsala, Sweden) and ImageQuant software (GE Healthcare, Chicago, IL, USA). For precise calculations, we used the amount of material in spots corresponding not only to m^5^dC but also to products of its degradation as dC (cytosine) and dT (thymine). The total m^5^C contents were calculated as R = m^5^dC/(m^5^dC + dC + dT) × 100. Each analysis was repeated four times.

### 4.8. Analysis of 8-oxo-dG Contents in DNA

DNA was dissolved in 200 µL of a buffer (pH 5.3) containing 40 mM sodium acetate and 0.1 mM ZnCl_2_, then mixed with nuclease P1 (Sigma-Aldrich, St. Louis, MO, USA) solution (30 µg), and incubated for 3 h at 37 °C. Then, 30 µL of 1 M Tris–HCl pH 8.0 and 5 µL of alkaline phosphatase (1.5 units) solution were added, followed by a 1 h incubation at 37 °C. DNA hydrolysate was purified using cut-off 10 kDa filter units. The 8-oxo-dG amount in DNA was determined using HPLC (Agilent Technologies 1260 Infinity, Santa. 25. Clara, CA, USA) with two detectors working in series: the 1260 Diode Array Detector and the Coulochem III Electrochemical Detector (ESA Inc., Chelmsford, MA, USA). The isocratic chromatography of DNA hydrolysate was performed using a solution of 50 mM CH_3_COONH_4_ at pH 5.3 and methanol (93:7). An analysis of dG for reference was performed at 260 nm. 8-oxo-dG was determined with the following electrochemical detection settings: guard cell +400 mV, detector 1: +130 mV (screening electrode), and detector 2: +350 mV (measuring electrode set on the 100 nA sensitivity).

### 4.9. Quantification of the Amount of m^5^C and 8-oxo-dG in Human Genomic DNA

A quantification of the genome amount of m^5^C and 8-oxo-dG in human DNA was conducted. The number of modified bases in DNA was calculated based on global genome composition. The amount of m^5^C (%) in pyrimidines in DNA was determined from TLC analysis with the formula “a”. The total number of m^5^C in the genome was calculated from the formula “b” (Figure 7). The input amount of guanosine was necessary to determine 8-oxo-dG contents. It was calculated from diode array detector (PAD) measurements using the Avogadro number NG = 6.02 × 10^20^ × b(mAU)/a(mAU) standard. The 8-oxo-dG nucleoside amount was estimated with the electrochemical detector N8-oxo-dG = 6.02 × 10^20^ × d(nA)/c(nA) standard. The total number of 8-oxo-dG = 623 × 10^6^ × N8-oxo-dG/NG (Figure 7) [78] was calculated.

### 4.10. Statistical Analysis

Microsoft Excel 2010 software with the Data Analysis package (14.0.7268.5000 - 32-bit) was used for the statistical analysis of all data. The data are the result of three independent experiments. The descriptive statistics function was used to generate the mean and SD, and the results were expressed in the error bars. A one-tailed t-test was used to calculate significant differences in R values for the tested samples compared with the control.

## 5. Conclusions

By analyses of cell viability, we showed that MF and Arg reduce TMZ cytotoxicity in glioblastoma cells. DNA methylation (m^5^C) content changes suggest the potential therapeutic effect of MF and Arg in cancer treatment. We have found the epigenetic mechanism of MF and Arg action on glioblastoma cells, which explains all cellular observations.

## Figures and Tables

**Figure 1 ijms-25-09460-f001:**
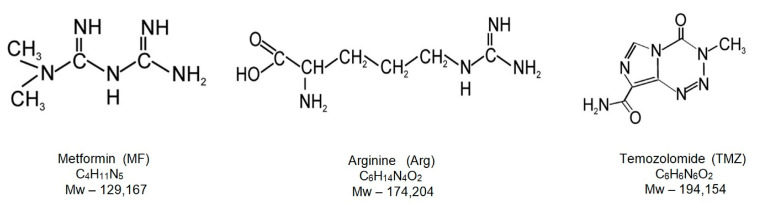
Structural formula of metformin, arginine and temozolomide.

**Figure 2 ijms-25-09460-f002:**
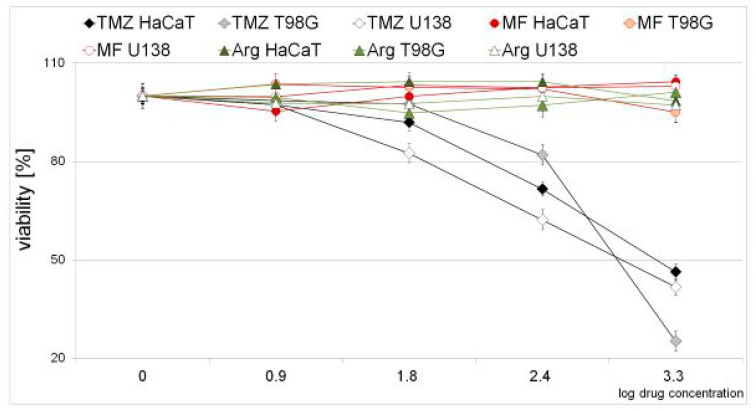
Cytotoxicity of temozolomide (TMZ), metformin (MF), and arginine (Arg). Effects on human keratinocyte (HaCaT), and glioblastoma (T98G, U138) cell lines were determined using the MTT test. The cells were treated with 1–2000 µM (0–3.3 on the logarithmic scale) for 24 h.

**Figure 3 ijms-25-09460-f003:**
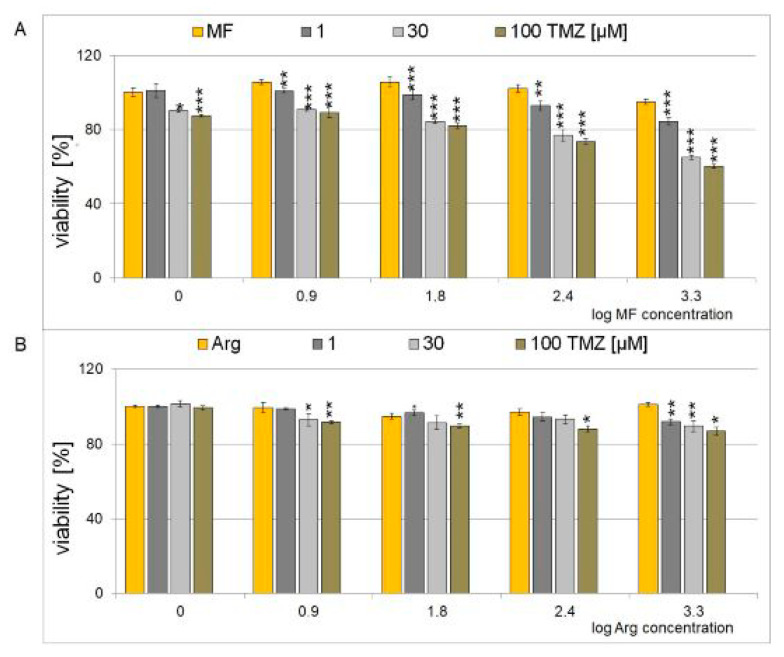
Effect of metformin (**A**) and arginine (**B**) on glioblastoma (T98G) cell line in the presence of TMZ. MF and Arg without TMZ are marked with orange. The analysis was performed after 24 h of incubation. The concentrations for metformin and arginine were in the range of 1–2000 µM (0–3.3 on the logarithmic scale) and the concentrations for TMZ: 1, 30, 100 µM. For control experiments, cells were treated with H_2_O only. Values are means ± SE from at least four experiments. Asterisks indicate a significant difference (* *p* < 0.05, ** *p* < 0.01, *** *p* < 0.001) from the control value.

**Figure 4 ijms-25-09460-f004:**
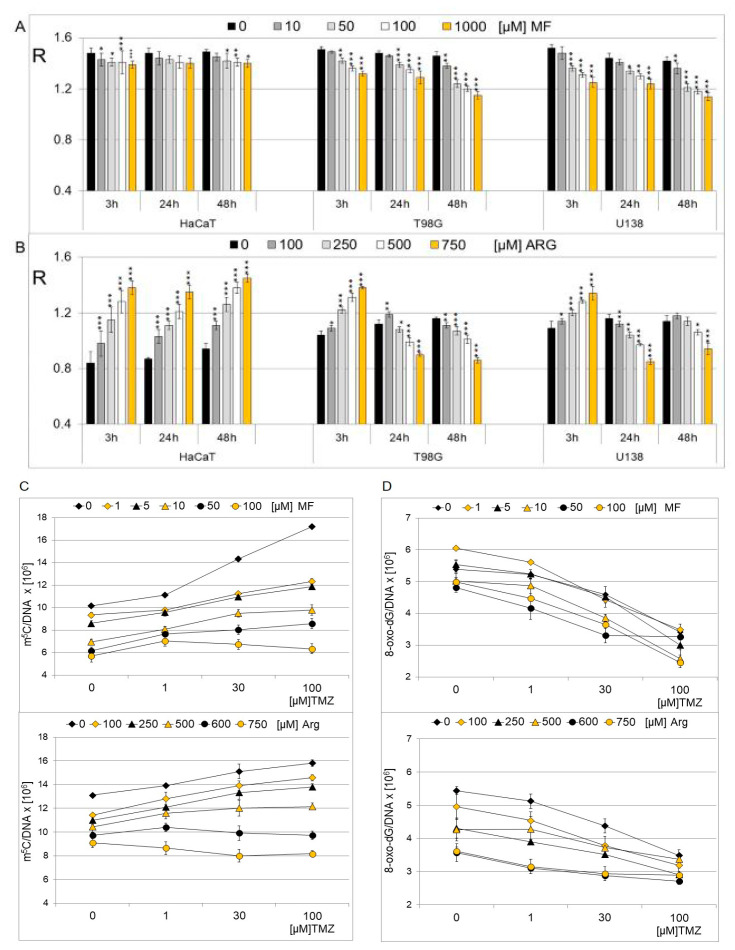
The effect of metformin (**A**) and arginine (**B**) on the total DNA (m^5^C) methylation level of the T98G, U138 and HaCaT cell lines. The analysis was performed after 3, 24, and 48 h of incubation at different MF and Arg concentrations (0–1000 µM, and 0–750 µM, respectively). Control cells (0) were treated with H_2_O only. Asterisks indicate a significant difference (* *p* < 0.05, ** *p* < 0.01, *** *p* < 0.001) from the control value. Quantification of m^5^C (**C**) and 8-oxo-dG (**D**) in DNA from T98G cell line after concomitant treatment with MF, Arg and TMZ. The analysis was performed after 24 h of incubation in given TMZ (0, 1, 30, 100 µM), MF (0, 1, 5, 10, 50, 100 µM), and Arg (0, 100, 250, 500, 600, 750 µM) concentrations. In control experiment for cells treated with TMZ, DMSO was used only. The values are means from three experiments.

**Figure 5 ijms-25-09460-f005:**
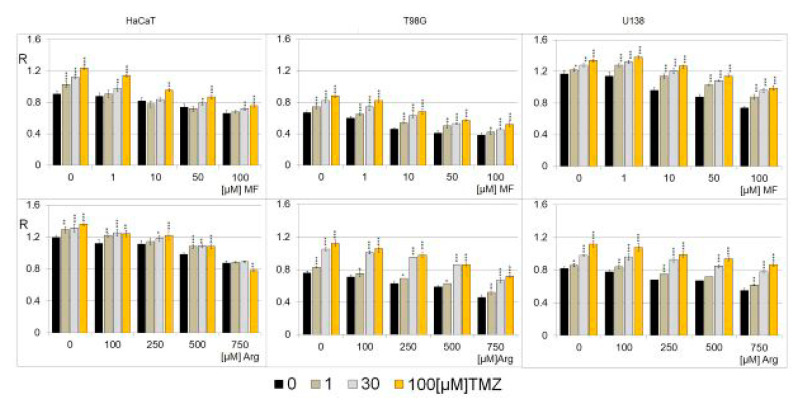
The effect of action of MF and Arg with TMZ on total DNA (m^5^C) methylation level in different cell lines (HaCaT, T98G, U138). The analysis was performed after 24 h of incubation in a given MF (0, 1, 10, 50, 100 μM), Arg (0, 100, 250, 500, 750 µM) and TMZ concentration (0, 1, 30, 100 μM). For control experiments, cells were treated with H_2_O only. The global m^5^C level values are means from three experiments. Asterisks indicate a significant difference (* *p* < 0.05, ** *p* < 0.01, *** *p* < 0.001) to the control value.

**Figure 6 ijms-25-09460-f006:**
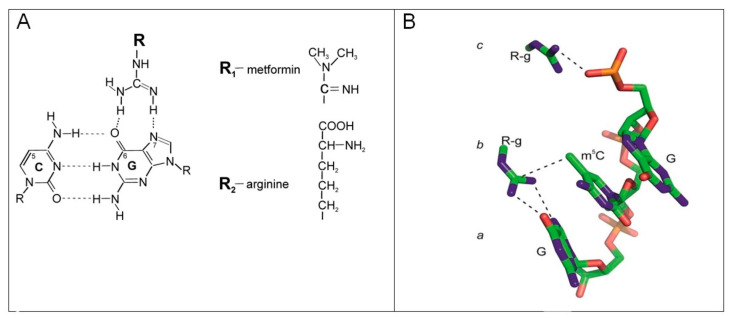
(**A**) Interaction of metformin and arginine with Hoogsteen edge of guanosine in DNA. Hydrogen bonds involve of O6 and N7 of guanosine. (**B**) Mechanism of recognition of guanidine group of metformin and arginine (R-g) within m^5^CpG of the DNA. It is based on the structure 4GZN from Protein Data Bank (PDB). It includes the following: a—Hydrogen bonds of 06 and N7 atoms with the 3′guanine, blocks methylation of 06 with TMZ. N7 involvement in H-bond formation diminishes the reactivity of carbon 8 of guanosine towards oxidation. b—Van der Waals interaction of guanidine moiety with a methyl group. c—Electrostatic interactions of guanidine moiety with a phosphate group.

**Figure 7 ijms-25-09460-f007:**
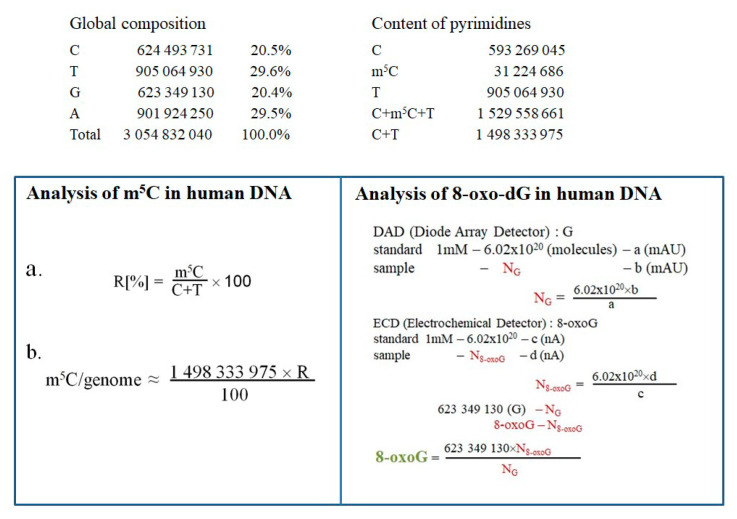
Calculations for determining the content of 5-methylcytosine (m^5^C) and 8-oxo-deoxyguanosine (8-oxo-dG) in the tested DNA samples. Data on the composition of human genomic DNA were taken from [78].

## Data Availability

The data presented in this work are available upon request from the corresponding author.

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
