# Peer review of "Mechanistic Insights on Metformin and Arginine Implementation as Repurposed Drugs in Glioblastoma Treatment"

_ijms, 2024, doi:10.3390/ijms25179460_

Round 1

Reviewer 1 Report

Comments and Suggestions for Authors

The article by Barciszewska et al. entitled ,, Mechanistic insights on metformin and arginine implementation as repurposed drugs in glioblastoma treatment’’ investigates the potential anti-GBM effect of two well know agents metformin and arginine. The trend of drugs repurposing is strong currently and the research seem interesting. However, I have certain concerns regarding this work that should be addressed before publication. My detailed comments are given below:

Major points

Introduction

1.                  Authors claim that arginine: ‘It is not essential in non-cancerous human cells but is crucial for the survival of cancer cells. Defective arginine synthesis is one of cancer’s most common metabolic weaknesses [53]. Cancer cells often alter  mitochondrial function, and arginine starvation has been shown to damage mitochondria, causing increased accumulation of ROS and subsequent genome instability [54-56]. Arginine supports several metabolic reactions, including the  synthesis of nitric oxide, polyamines, glutamine, and proline, all of which are important  cell growth and survival regulators’ etc. As far as I understand, arginine deficiency should be favoring for cancer cell elimination. Please correct me if I am wrong. Otherwise, please provide better explanation why use arginine as anti-GBM supporting agent?

Results

2.                  My main issue here is: why the results are not statistically analyzed? Authors claim to perform the experiments in 3 (or even 4 for the MTT) repeats. It should not be difficult to add the statistical significance markers in the graphs.

3.                  Moreover, section ‘2.2. Cytotoxic effect of a combination of metformin, and arginine with temozolomide in neoplastic cell line’. Authors proceed to analyze only the T98G cell line. What does it mean neoplastic in this context? Isn’t U138 neoplastic? Please explain.

4.                  Why in the caption of Figure 6 the DMSO is indicated to be used as vehicle in control cells, whereas in the rest of the experiments it seems like H2O served as diluting agent?

Discussion

5.                  I think that Figures 7 and 8 should be removed from the Discussion section. In my opinion they should be incorporated into the Results section of the manuscript. I would suggest to create a multi-panel Figures 4 and 6 (or wherever Authors think it suits better to explain the mechanisms of action of tested compounds). In my opinion it would provide clearer overview of the whole manuscript layout.

Materials and Methods

6.                  Subsection 4.2. Cell line and culture conditions: please include the info about the solubility of used compounds to explain why H2O/DMSO was used as vehicle.

7.                  Subsection 4.3. Cell viability/proliferation assay: I would not call MTT assay a proliferation assay as viability does not equal proliferation. Please stick with the cell viability assay.

8.                  Subsection 4.6. Isolation of DNA, it is stated: ‘DNA from tissue samples was extracted with a Genomic Mini kit. Shortly, tissue samples were incubated with proteinase K first and then with RNase A.’ Where was the tissue analyzed? It looks like only cell lines were studied in the manuscript?

9.                  Subsection 4.10. Statistical analysis: please provide proper statistical analysis. With 3 experimental repeats is should not be any problem.

Conclusions

10.              If possible, please provide a short Conclusions section summarizing the main points of the manuscript. Since the study is not very straightforward in the interpretation, conclusions should help the reader with the perception of the results.

Minor points

Please read the manuscript carefully to identify certain errors, e.g. typos such as in line 197: (T98G, U138, ); 4.6 Isolation of DNA, should rather be: Isolation of the DNA.

Comments on the Quality of English Language

Minor corrections are needed

Reviewer 2 Report

Comments and Suggestions for Authors

In this manuscript, Barciszewska et al. explore the repurposing of metformin and arginine for glioblastoma treatment. These drugs, known for their safety profiles, were found to decrease DNA methylation in glioblastoma cells. Combined with temozolomide, they increased DNA methylation and reduced oxidative stress. The study suggests that metformin and arginine, alone or with temozolomide, could be effective in treating glioblastoma at various stages.

The paper is of interest however the conclusion are not sufficiently supported by the data at the current stage of the manuscript.

Here is a list of suggestion to improve the quality of the manuscript:

1)      I might have missed something but why in T98G cell line in Fig. 2 the viability is dropping to 20% with 3.3 micromolar and then in Fig. 3 with 100micromolar the viability is ~80%?

2)      MTT is a good test for viability but when it comes to chemotherapy sensibility it is usually preferred to apply a proliferation assay (96 hours usually with 24h sampling). This is even more important when they compared condion where some of the drugs used have an impact on the cell metabolism (Metformin). The author should replicate some of the main results making the proliferation assay.  

3)      The authors mention the “synergetic action of all the drugs” but this statement must be supported by making Combination Index measurement.

Minor:

1)      There are couple of format mistakes along the manuscript (line 82 for example). Please fix them.

Round 2

Reviewer 1 Report

Comments and Suggestions for Authors

I think the manuscript has been improved in comparison to its first version. However, it is very difficult to track the changes introduced by Authors, since they did not make an effort to mark all the alterations.

I appreciate the explanation about the use of arginine, but I have no idea if Authors introduced at least brief mention justifying this choice in the main text of their manuscript.

Also, Authors admit that U138 is neoplastic too, but in the manuscript the title of the subsection is still:  ‘Cytotoxic effect of a combination of metformin, and arginine with temozolomide in neoplastic
cell line.’, suggesting that only T98 cells are neoplastic. Moreover, I was expecting at least brief justification of the selection of T98 in this experiment; should be explained in the manuscript.

Still no info about the solubility of tested compounds in the Materials and methods? Placing this info only in figure captions is not enough.

Lastly, I am wondering why Authors chose to use t-test for statistical analysis. T-test is usually used when comparing up to two groups. Here, multiple groups are analyzed. In my opinion ANOVA would be more appropriate. However, maybe Authors have their point, I am just curious about the choice of this test.

Comments on the Quality of English Language

Some editing is necessary.

Reviewer 2 Report

Comments and Suggestions for Authors

The authors have thoroughly addressed all the suggestions, resulting in a significantly improved manuscript that is now well-suited for publication.

Best,
